# The Association of Low Skeletal Muscle Mass with Complex Distal Radius Fracture

**DOI:** 10.3390/jcm11195581

**Published:** 2022-09-22

**Authors:** Chi-Hoon Oh, Junhyun Kim, Junhan Kim, Siyeong Yoon, Younghoon Jung, Hyun Il Lee, Junwon Choi, Soonchul Lee, Soo-Hong Han

**Affiliations:** 1Department of Orthopedic Surgery, CHA Bundang Medical Center, CHA University School of Medicine, 335 Pangyo-ro, Bundang-gu, Seongnam-si 13488, Gyeonggi-do, Korea; 2Department of Orthopedic Surgery, Ilsan Paik Hospital, Inje University, 170, Juhwa-ro, Ilsangeo-gu, Goyang-si 10380, Gyeonggi-do, Korea; 3Department of Molecular Science and Technology, Ajou University, Suwon-si 16499, Gyeonggi-do, Korea

**Keywords:** distal radius fracture, sarcopenia, osteoporosis, low skeletal muscle mass

## Abstract

Objectives: Sarcopenia is a skeletal muscle loss disease with adverse outcomes, including falls, mortality, and cardiovascular disease (CVD) in older patients. Distal radius fractures (DRF), common in older people, are strongly related to falls. We aimed to investigate the correlation between DRF and low skeletal muscle mass, which strongly correlated to sarcopenia. Methods: We performed a retrospective review of data from patients diagnosed with or without DRF in our institute between 2015 and 2020. Finally, after propensity score matching, data from 115 patients with and 115 patients without DRF were used for analyses. Multivariate logistic regression analysis was performed for sex, body mass index (BMI), the presence of low skeletal muscle mass, bone quality measured by dual-energy X-ray absorptiometry (DXA), and comorbidities (diabetes mellitus, CVD). Results: We found that female sex (odds ratio = 3.435, *p* = 0.015), CVD (odds ratio = 5.431, *p* < 0.001) and low skeletal muscle mass (odds ratio = 8.062, *p* = 0.001) were significant predictors for DRF. BMI and osteoporosis were not statistically significantly related to DRF. Conclusions: Women with low skeletal muscle mass and CVD may be more responsible for DRF than osteoporosis.

## 1. Introduction

Sarcopenia is a Greek term signifying “poverty of the flesh”; which was first described in the 1980s as an age-related decline in skeletal muscle mass affecting mobility, nutritional status, and independence [1]. Since its evolution, the definition has focused on muscle function, usually defined by muscle strength, muscle power, or physical performance; these have consistently been shown to be more powerful predictors of clinically relevant outcomes than muscle mass alone [2,3,4]. In 2010, the European Working Group on Sarcopenia in Older People (EWGSOP) proposed a diagnostic algorithm for sarcopenia; in which both muscle quantity and quality are basic requirements [5]. However, Asians bear different features than Europeans; namely a relatively higher adiposity, smaller body size, and less mechanized and more physically active lifestyles [6]. AWGS 2019 retained the previous definition of sarcopenia; however, it revised the diagnostic algorithm, protocols, and some criteria as follows:

In order to diagnose sarcopenia in the primary health care or community preventive services setting, calf circumference (men: <34 cm, women: <33 cm) or a SARC-F questionnaire (≥4), or SARC-F administration combined with CC measurement (SARC-CalF ≥11) was presented at the start of the study. Then, muscle strength measurement or physical performance was measured by the process of assessment; this measured handgrip strength (men: <28 kg, women: <18 kg) and a physical performance measure 5-time chair stand test (≥12 s), before proceeding to the final diagnostic process. In the final diagnosis step, the following three were considered:

Low muscle strength as handgrip strength of <28 kg for men and <18 kg for women;Low physical performance as a 6-m walk of <1.0 m/s or short physical performance battery score of ≤9, or a 5-time chair stand test of ≥12 s;Low appendicular skeletal muscle mass (ASM) as a <7.0 kg/m^2^ in men and <5.4 kg/m^2^ in women by dual-energy X-ray absorptiometry (DXA); or <7.0 kg/m^2^ in men and <5.7 kg/m^2^ in women by bioimpedance;

At this time, sarcopenia was defined as low ASM + low muscle strength or low physical performance; and severe sarcopenia was defined as low ASM + low muscle strength and low physical performance [7].

Distal radius fractures (DRF) are among the most common fractures (18%) in patients aged 65 years and older [8]. These fractures have an annual incidence of >640,000 in the United States [9] and cost approximately $170 million in 2007 from Medicare claims alone [10]. In women, the incidence has increased exponentially since the age of 50; and falling while walking outside was the most common mechanism of injury [11]. Accordingly, an increased risk of falls is expected to be a risk factor for distal radial fractures [12]. DRF is one of the most common fractures caused by osteoporosis, and many studies have reported that low-energy trauma-induced DRF is associated with osteoporosis [13,14,15,16]. Distal radial fractures occur on average 15 years earlier than hip fractures [17]. Therefore, the risk of DRF may be caused by a combination of maintained physical activity for walking and falls due to subtle declines in physical performance [18].

According to a meta-analysis that included 10 cohort studies, sarcopenia is associated with falls [19]. Sarcopenia is characterized by a decrease in fast muscle mass and the number of motor neurons; with a significant negative impact on muscle mass, strength, and physical performance [20,21]. This may affect physical abilities, including gait speed, endurance, and balance; all of which may be associated with falls [22]. However, there are some studies on the relationship between DRF and sarcopenia; and this relationship is controversial [23,24,25,26].

Here, we investigated the relationship between DRF and low skeletal muscle mass; being strongly related to sarcopenia.

## 2. Materials and Methods

### 2.1. Study Population

This was a retrospective comparative study of patients with DRF who visited the CHA Bundang Medical Center between January 2015 and December 2020 under approval of the CHA Bundang Medical Center Institutional Review Board (CHAMC 2022-07-024-002). All the patients with DRF underwent open reduction and internal fixation surgery using a 2.4 mm variable-angle LCP two-column volar distal radius plate (Synthes, Oberdorf, Switzerland). Indications for surgery included the following Lafontaine criteria [27]: dorsal angulation greater than 20 degrees, dorsal comminution, intraarticular radiocarpal fracture, associated ulnar fracture, and an age of more than 60 years; three or more of the five parameters. The exclusion criteria were patients >80 years of age; who had severe comorbidities that influenced musculoskeletal activities (e.g., cancer, cerebrovascular accident, bed ridden status); and patients who had multiple organ injury and high-energy trauma.

Data on individuals who underwent DXA for a health check-up without any specific history of osteoporotic fracture (e.g., vertebral, hip, distal radius), which measures bone mineral density (BMD) and ASM, were used as a patient collective for the control group.

Demographic data, including age, sex, height, and weight (along with the BMI calculated in kg/m^2^) were collected. Subsequently, propensity score-matching was performed for age. Finally, the participants were divided into two groups based on whether they had been diagnosed with DRF (*n* = 115 in each group) (Figure 1).

Variables included sex; age; BMI; T-score from BMD; the presence of osteoporosis; and comorbidities, including diabetes mellitus (DM) and cardiovascular disease (CVD). This study’s definition of cardiovascular disease included hypertension, coronary heart disease, peripheral arterial disease, and aortic disease.

BMD was measured at both the lumbar spine and one proximal femoral area (Lunar Prodigy Advance; GE Lunar, Madison, WI, USA). According to the World Health Organization criteria, osteopenia was defined as a T-score standard deviation (SD) −2.5< T <−1.0; and osteoporosis was defined as T ≤ −2.5. Appendicular lean body mass without fat and bone mass were measured separately in the upper and lower extremities through DXA. Appendicular lean mass corresponds approximately to the ASM [28,29]. The ASM was divided by the height squared as a relative value for comparison. Adjusted criteria from AWGS 2019 for considering Asian low muscle volume were employed to diagnose low skeletal muscle mass (men <7.0 kg/m^2^; women <5.4 kg/m^2^).

### 2.2. Statistical Analyses

Data manipulation and statistical analyses were performed using the R software (version 3.6.3; The R Foundation for Statistical Computing, Vienna, Austria; http://www.r-project.org/, accessed on 15 July 2022). Continuous normally distributed data are presented as mean and SD. Statistical significance was set at *p* < 0.05. A Pearson’s chi-squared test for categorical variables and a Student’s *t*-test for continuous variables were used to compare the groups. Univariate analysis was performed to determine factors associated with DRF. Multivariate logistic regression analysis was used, adjusting for factors and covariates, to confirm the simultaneous effect of multiple factors associated with DRF. The results are presented as odds ratios (OR) with 95% confidence intervals (CIs) and Akaike information criteria (AIC) values, which estimate the quality of each model. We applied a propensity score matching method to reduce the possible bias originating from the difference in the patients’ demographic characteristics. The patients with DRF were matched with the non-DRF group on the basis of a greedy algorithm of nearest neighbor matching at a 1:1 fixed ratio. Age was considered as a predefined covariate. We obtained the same number of individuals for the DRF and non-DRF groups.

## 3. Results

### 3.1. Demographics

The initial data showed demographic differences between the two groups in terms of age, osteoporosis, CVD, and the presence of low skeletal muscle mass (Table 1). However, after propensity score matching, no statistically significant differences were identified between the groups in terms of age and osteoporosis (Table 2). Overall, 115 patients with DRF and 115 controls were enrolled before statistical comparison. An additional analysis was performed between the two groups on the skeletal muscle mass according to gender (Table 3).

### 3.2. Multivariate Logistic Regression Analysis of Factors Associated with DRF

The association between low skeletal muscle mass and DRF according to the multistep adjustments is shown in Table 4. The male sex variable was adjusted as a reference. BMI, the presence of low skeletal muscle mass, the T-score of BMD, the presence of osteoporosis, DM, and CVD were adjusted in multiple steps. Women (i.e., the female sex) had no statistically significant difference in model 1; however, after adjusting for all the variables, there was a significant association with DRF in model 4 (OR 3.435, *p* = 0.015). The patients with a low skeletal muscle mass were significantly associated with DRF in all the models. After adjusting for all the variables, the OR of low skeletal muscle mass was 8.062 (*p* = 0.001) in model 4. CVD was also statistically related to DRF; the OR of CVD was 5.431 (*p* < 0.001) in model 4. The presence of osteoporosis, T-score, and BMI were not related to DRF in model 4.

## 4. Discussion

In our study, after propensity score matching was adjusted for age, DRF was associated with the female sex (OR 3.435, *p* = 0.015), patients with a low skeletal muscle mass (OR 8.062, *p* = 0.001), and CVD (OR 5.431, *p* < 0.001), compared with the control group.

In this study, the female sex was associated with DRF. Considering that the age of the subjects in this study was on average above the age of menopause, this was consistent with results from previous literature. DRFs have been reported to occur more frequently in post-menopausal women in studies conducted in the past 20 years compared with men starting at a later age [30,31,32].

Few studies have investigated the direct relationship between sarcopenia and DRF. In addition, the results are controversial. According to Caliskan et al. [23], a cross-sectional study involving 28 controls without fractures and 27 patients with DRF who visited geriatric outpatient clinics found that both groups’ rates of sarcopenia were comparable. Lee et al. [24] reported that patients with DRF did not have significantly higher sarcopenia rates; however, BMD was significantly lower in patients with DRF than in controls. Recently, Artiaco et al. [26] analyzed sarcopenia in DRFs through a systematic review. This report concluded that approximately 30% of patients older than 50 years with DRFs suffered from sarcopenia. In addition, we found that low skeletal muscle mass was associated with DRF (OR 8.062, *p* = 0.001). In this study, low skeletal muscle mass was related to DRF; however, there were conflicting results in previous literature. Hence, further research is needed on this matter.

In our study, DRF patients had the characteristics of CVD. A multicenter cross-sectional study reported that hypertension was associated with an increased risk of sarcopenia at least twice [33]. Moreover, sarcopenia has been reported to be significantly associated with metabolic syndrome (OR 3.073, CI 2.009–4.701, *p* < 0.001) [34]. Several mechanisms, including lack of physical activity, insulin resistance, inflammation, and myokine, can affect the association between sarcopenia and metabolic syndrome [35,36]. Considering the previous literature and the results of this study together, the relationship between CVD and DRF is thought to be related to the individual’s health status and physical activity.

This study showed that osteoporosis was not associated with DRF (OR 0.983, *p* = 0.947). Since DRF is considered a fracture type that defines “severe or established osteoporosis” [37], the results of this study do not mean that the two factors are actually unrelated, but that osteoporosis is common in the older population; it may not affect the difference between the control group and the patient group. In this study, there was no difference in the presence of osteoporosis between the patient and control groups. Sosa et al. [38] reported no difference in the risk of Colles’ fracture in 486 postmenopausal Spanish women compared with the control group. The average age of the patients in this study was 64.0 ± 8.3 years. Nordvall et al. [39] reported no difference in the T-scores compared with the control group; however, the patients aged over 64 years had a history of falling more often than the corresponding controls (*p* = 0.01). Additionally, in this study, patients aged 45–64 years had lower T-scores (*p* = 0.09); however, there was no difference in falls. These results suggest that not only osteoporosis, but also the effects of falls, should be considered as risk factors for DRF in relatively older patients. As an additional point to consider, the World Health Organization criteria define osteoporosis as low BMD, with a T-score of ≤−2.5 in the spine and neck of the femur [40]. The fact that DXA does not reflect the actual bone density of the distal radius may have affected our results.

A limitation of this study is that only ASM from DXA was used to diagnose low skeletal muscle mass. There were no data on handgrip strength or physical performance; therefore, this study did not meet the sarcopenia criteria corresponding to the AWGS 2019. As this study was conducted only with patients with DRF, the measurement of hand grip strength was limited due to pain and immobilization. Moreover, according to a recent scoping review of DRF with sarcopenia, only 3 out of 13 studies defined sarcopenia based on AWGS or EWGSOP criteria that consider age-related loss of muscle mass, low muscle strength, and/or low physical performance [41]. Second, we only included cases of DRF who underwent surgery in this paper because we hypothesized that low skeletal muscle mass would be related to complex intra-articular fractures. Hence, we cannot determine the relationship between low skeletal muscle mass and DRF without surgery. In future studies, it will also be necessary to include patients with DRF who did not undergo surgery.

## 5. Conclusions

Although DRF is highly associated with osteoporosis in the literature, considering the above, factors related to general activity, such as low skeletal muscle mass and CVD, may be more important factors in causing DRF in women. In addition, research related to sarcopenia and DRF has been actively conducted. Moreover, future studies on sarcopenia will be of great assistance in the management of DRF, which is prevalent in society.

## Figures and Tables

**Figure 1 jcm-11-05581-f001:**
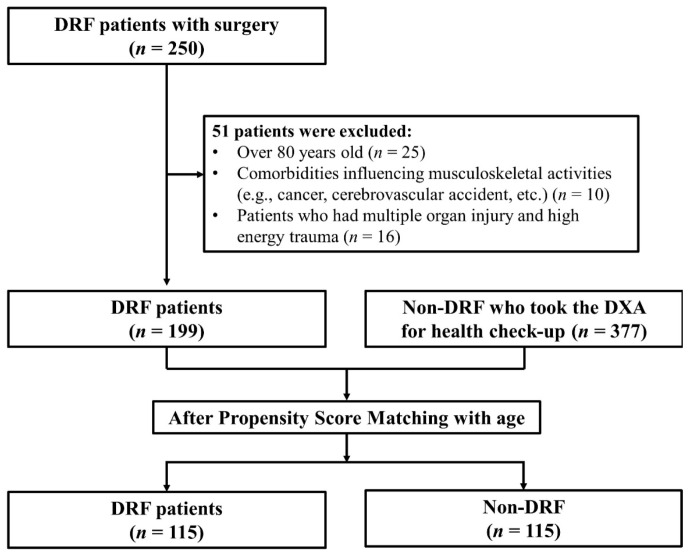
DRF: distal radius fracture, DXA: dual-energy X-ray absorptiometry.

**Table 1 jcm-11-05581-t001:** Univariate analysis of factors associated with DRF.

Variable	DRF (*n* = 250)	Non-DRF (*n* = 368)	*p*-Value
**Sex**			0.154 ^†^
Men	45 (18.00%)	85 (23.10%)	
Women	205 (82.00%)	283 (76.90%)	
**Age**	65.38 ± 9.19	71.45 ± 9.22	<0.001 *
**BMI (kg/m^2^)**	24.00 ± 3.31	23.64 ± 3.59	0.211 *
**T-score**			
Spine	−1.87 ± 1.33	−1.88 ± 1.03	0.897 *
Femur	−1.49 ± 1.11	−1.67 ± 1.01	0.065 *
**Osteoporosis**			0.009 ^†^
Normal	28 (17.50%)	31 (8.42%)	
Osteopenia	70 (43.75%)	185 (50.27%)	
Osteoporosis	62 (38.75%)	152 (41.30%)	
**Comorbidity**			
DM	32 (12.85%)	67 (18.21%)	0.096 ^†^
CVD	75 (30.00%)	50 (13.59%)	<0.001 ^†^
**Low skeletal muscle mass**	33 (25.58%)	26 (7.07%)	<0.001 *

* *t*-test, ^†^ Pearson’s chi-square test; DRF: distal radius fracture, DM: Diabetes mellitus, CVD: Cardiovascular disease.

**Table 2 jcm-11-05581-t002:** Univariate analysis of factors associated with DRF after adjusting Age.

Variable	DRF (*n* = 115)	Non-DRF (*n* = 115)	*p*-Value
**Sex**			0.223 ^†^
Men	16 (13.91%)	24 (20.87%)	
Women	99 (86.09%)	91 (79.13%)	
**Age**	67.47 ± 8.30	67.48 ± 8.32	0.994 *
**BMI (kg/m^2^)**	24.25 ± 3.44	23.76 ± 3.51	0.283 *
**T-score**			
Spine	−1.83 ± 1.42	−1.84 ± 1.14	0.959 *
Femur	−1.57 ± 1.16	−1.78 ± 1.01	0.140 *
**Osteoporosis**			0.476 ^†^
Normal	18 (15.65%)	12 (10.43%)	
Osteopenia	54 (46.96%)	55 (47.83%)	
Osteoporosis	43 (37.39%)	48 (41.74%)	
**Comorbidity**			
DM	16 (14.04%)	19 (16.52%)	0.734 ^†^
CVD	42 (36.52%)	13 (11.30%)	<0.001 ^†^
**Low skeletal muscle mass**	29 (25.22%)	4 (3.48%)	<0.001 *

* *t*-test, ^†^ Pearson’s chi-square test; DRF: distal radius fracture, DM: Diabetes mellitus, CVD: Cardiovascular disease.

**Table 3 jcm-11-05581-t003:** Comparison of skeletal muscle mass by DRF.

	DRF (%)	Non-DRF (%)	*p*-Value
**Total skeletal muscle mass**	115 (50%)	115 (50%)	<0.001 ^†^
Normal	86 (74.78%)	111 (96.52%)	
Low	29 (25.22%)	4 (3.48%)	
**Men skeletal muscle mass**	16 (40%)	24 (60%)	0.005 ^†^
Normal	10 (62.50%)	24 (100.00%)	
Low	6 (37.50%)	0 (0.0%)	
**Women skeletal muscle mass**	99 (52.11%)	91 (47.89%)	<0.001 ^†^
Normal	76 (76.77%)	87 (95.60%)	
Low	23 (23.23%)	4 (4.40%)	

^†^ Pearson’s chi-squared test, DRF: distal radius fracture.

**Table 4 jcm-11-05581-t004:** Multivariate logistic regression analysis of factors associated with DRF.

	Model 1		Model 2		Model 3		Model 4	
	AIC = 294.82		AIC = 296.97		AIC = 258.98		AIC = 242.46	
Variable	OR (95% CI)	*p*-Value	OR (95% CI)	*p*-Value	OR (95% CI)	*p*-Value	OR (95% CI)	*p*-Value
**Sex (Ref. Men)**	1.794 (0.856–3.895)	0.128	2.097 (0.964–4.768)	0.068	3.181 (1.298–8.367)	0.014	3.435 (1.311–9.751)	0.015
**BMI**	1.098 (1.013–1.194)	0.025	1.094 (1.009–1.19)	0.033	1.086 (0.991–1.194)	0.079	1.077 (0.977–1.19)	0.138
**Low skeletal muscle mass**	12.273 (4.436–43.898)	<0.001	12.222 (4.384–44.03)	<0.001	9.819 (3.364–36.682)	<0.001	8.062 (2.659–30.916)	0.001
**T-score**								
Spine			0.602 (0.236–1.492)	0.277	0.562 (0.191–1.578)	0.28	0.514 (0.162–1.562)	0.246
Femur			0.52 (0.197–1.331)	0.177	0.392 (0.127–1.144)	0.092	0.394 (0.118–1.253)	0.119
**Osteoporosis (Ref. Normal)**								
Osteopenia					0.917 (0.51–1.642)	0.771	1.107 (0.59–2.083)	0.751
Osteoporosis					0.973 (0.606–1.569)	0.909	0.983 (0.595–1.637)	0.947
**Comorbidity**								
DM							0.708 (0.276–1.745)	0.458
CVD							5.431 (2.453–13.031)	<0.001

OR: odds ratio, CI: confidence interval, DM: Diabetes mellitus, CVD: Cardiovascular disease, DRF: distal radius fracture, Ref.: reference.

## Data Availability

The data presented in this study are available on request from the corresponding author. The data are not publicly available in accordance with national data safety guidelines.

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
