# Peer review of "The Association of Low Skeletal Muscle Mass with Complex Distal Radius Fracture"

_jcm, 2022, doi:10.3390/jcm11195581_

Round 1

Reviewer 1 Report

General comments:

The authors are presenting a retrospective cohort study, comparing DRF with a matched control group to determine risk factors for DRF. Reduced muscle mass and the consecutive loss of strength is an interesting topic in discussing risk factors for DRF. The study is methodologically well conducted, besides the following issue:

According to the AWGS criteria, muscle strength or physical performance was not assessed in the present study. By definition, it is not eligible to diagnose sarcopenia only using ALM. Therefore, you actually assessed the association between skeletal muscle mass and DRF, because functional parameters were not assessed. Thus, you might consider rephrasing the title and purpose of the study accordingly.

Introduction section, M&M, and results section are very well written, besides some minor comments below.

It would also be interesting, which Type of DRF according to the AO classification was included in the present study. I am especially asking for this detail, because you have only included patients who received surgical treatment for DRF. In other words, what were your indications for surgical treatment for DRF? Why were conservatively treated patients with DRF not eligible for inclusion?

I would strongly recommend revising the discussion section. In my opinion, it could be shortened and be more focused. Please only discuss results, which also have been mentioned in the results section.

Moreover, I recommend to completely revise the conclusion section. Focus on the main findings of your study.

Please make sure, that you use the font according to the MDPI template throughout the whole manuscript.

Abstract:

Line 27: Why do you mention cerebrovascular accident and cancer in this section. In Line 86 you mention that you have excluded patients with cancer. Cerebrovascular accident is neither mentioned in the M&M nor the Results section. Please clarify.

Line 30: Omitting “in these two groups” would render this sentence more comprehensible.

Line 31: Please rephrase this sentence.

Introduction:

The Introduction Section is well written and gives a good introduction to the topic and leads the reader to the study’s research question.

Line 46 – 49: In my opinion, it would be fine to simply state the current definition of AWGS (2019). This section can be omitted to keep the paragraph focused.

Line 55: a BMI of < 7.0 or < 5.4 cannot be the correct figure. In Ref. 7 these figures are referred to the ALM measurement.

Line 49-56: I would appreciate, if you could emphasize the 3 diagnostic groups “Muscle Strenght”, “Physical Performance” and “ASM” more. This would make this section more comprehensible.

Materials and Methods:

Thank you for providing the patient flowchart in Figure 1. It is a nice overview, how the number of patients was created.

The statistical methods are appropriate for this study design.

Please include in the M&M Section, that you have assessed comorbidities. Moreover, please state, how you defined cardiovascular disease (arterial hypertension?,  coronary heart disease?, peripheral/cerebral arterial disease?)

L 101: As you have assigned your participants to normal BMD, osteopenia and osteoporosis in Table 1, please specify “osteopenia” in this section.

Line 103: Please introduce ALM before using it as abbreviation. Line 89 would be appropriate for doing so.

Results:

Table 1: The mean age in Non-DRF was stated as 71.45 ± 9.22. Did you also apply a max. age of 80 years for this group? Statistically, there must be some patients older than 80 with this mean and SD value.

Discussion:

Line 162: I cannot find the information in the results section, where age was assessed as risk factor for DRF. Please clarify.

Line 163-165: This sentence is rather confusion. Each variable resulting in a significant result in the multiple regression analysis should be stated equivalently. Please rephrase this sentence accordingly.

Line 166-172: Please state the information from this reference correctly. 86 should be rounded correctly to 87 years. “the same as the world's highest life expectancy in 2012, 169 with 57% of these being over 90 years” –> this sentence was reported in a confusing way. Please correct.

Line 173-174: I cannot find any statistical testing assessing a difference or correlation between age and sarcopenia in the results section.

Line 184: As mentioned in the general comments, neither muscle strength nor physical performance were included as parameters in the present study. Thus, you cannot predict an association, because you did not assess one of the two parameters.

Line 196: “Coliskan et al concluded in their cross-sectional study including 27 patients with DRF and 28 controls without fractures that sacopenia…” would make this sentence more comprehensible.

Line 214: Do you mean that other studies investigation the association between sarcopenia and DRF did include younger patients. Please clarify.

Line 215: The precise definition of CVD must be included in the M&M Section. If available, you could include a detailed list of precise CVD in the results section.

Line 228-231: I consider this sentence as very important in the discussion section. You may rephrase it in a way that DRF is considered as fracture type, which defines a “Severe or established osteoporosis”.  Please see:

Kanis JA on behalf of the World Health Organization Scientific Group. Technical Report. World Health Organization Collaborating Centre for Metabolic Bone Diseases, University of Sheffield; UK: 2007. 2007. Assessment of osteoporosis at the primary health-care level

Line 249: Except that the dominant hand is normally slightly stronger, assessing the hand not affected by DRF could have been an option.

Conclusion:

Line 258-259: It is hard to understand, what you want to express with this sentence. Please rephrase or omit.

Line 259: why don’t you include female gender in this sentence as significant outcome of your multivariate regression analysis?

Author Response

We sincerely thank you for your detailed and accurate review of our paper.

General comments:

The authors are presenting a retrospective cohort study, comparing DRF with a matched control group to determine risk factors for DRF. Reduced muscle mass and the consecutive loss of strength is an interesting topic in discussing risk factors for DRF. The study is methodologically well conducted, besides the following issue:

According to the AWGS criteria, muscle strength or physical performance was not assessed in the present study. By definition, it is not eligible to diagnose sarcopenia only using ALM. Therefore, you actually assessed the association between skeletal muscle mass and DRF, because functional parameters were not assessed. Thus, you might consider rephrasing the title and purpose of the study accordingly.

->Thank you for your careful reading, I have edited the title according to your comments. The new title is “The Association of Low Skeletal Muscle Mass with Complex Distal Radius Fracture”.

Introduction section, M&M, and results section are very well written, besides some minor comments below.

It would also be interesting, which Type of DRF according to the AO classification was included in the present study. I am especially asking for this detail, because you have only included patients who received surgical treatment for DRF. In other words, what were your indications for surgical treatment for DRF? Why were conservatively treated patients with DRF not eligible for inclusion?

->Thank you for your great comment. It would be great if we could analyse the distal radius fracture without surgery also.

However, first, to measure skeletal muscle mass, dual energy x-ray absorptiometry (DXA) including body composition test should be performed. Unfortunately, DXA was only performed on patients who underwent surgery in this hospital. Because of the retrospective nature of this study, we could not get the data.

Second, we hypothesized that patients with low skeletal muscle mass would have complex and unstable DRF that usually required surgical treatment. So, we focused to analyse the distal radius fracture with surgery.

We added your valuable comment as the limitation of this study. Please find below.

Second, we only included the DRF with surgery in this paper because we hypothesized that low skeletal muscle mass would be related with the complex intra-articular fracture. Hence, we cannot determine the relationship between low skeletal muscle mass and DRF without surgery. In the future study, it will be necessary to include the patient of DRF without surgery also.

Next, regarding the AO classification, we found that AO classification could not give the information of fracture stability. Instead of AO classification, we used Lafontaine criteria to decide the surgical treatment. Indications for surgery included the following Lafontaine criteria: dorsal angulation greater than 20 degrees, dorsal comminution, intraarticular radiocarpal fracture, associated ulnar fracture, age of more than 60 years; three or more of the five parameters.

Ref: Lafontaine, M., D. Hardy, and P. Delince. "Stability Assessment of Distal Radius Fractures." Injury 20, no. 4 (1989): 208-10.

I would strongly recommend revising the discussion section. In my opinion, it could be shortened and be more focused. Please only discuss results, which also have been mentioned in the results section.

->The discussion section has been further reduced and summarized based on the comments below. To focus on the results, those that were less relevant to the findings of this study, such as age, were omitted. The number of words in the discussion was 1195 before and 935 now

Moreover, I recommend to completely revise the conclusion section. Focus on the main findings of your study.

->The conclusion has been revised to focus more on the main finding.

Please make sure, that you use the font according to the MDPI template throughout the whole manuscript.

->Thank you for your comment. We revised using MDPI template and, also shortened the discussion.

Abstract:

Line 27: Why do you mention cerebrovascular accident and cancer in this section

->Thanks for your comment. It was our typo mistake. Sorry for the confusion. Some errors were excluded. Please find the revised manuscript below.

Multivariate logistic regression analysis was performed for sex, body mass index (BMI), the presence of low skeletal muscle mass, bone quality measured by dual-energy X-ray absorptiometry (DXA), and comorbidities (diabetes mellitus, CVD).

In Line 86 you mention that you have excluded patients with cancer. Cerebrovascular accident is neither mentioned in the M&M nor the Results section. Please clarify

->Thanks for your comment. Again, it was our mistake. Some errors were excluded and we mentioned Cerebrovascular accident in M&M

Line 30: Omitting “in these two groups” would render this sentence more comprehensible.

->We omitted it as you comment us. Thank you for comment

Line 31: Please rephrase this sentence.

->We rephrased this sentence. Please find the revised manuscript below.

In older, active individuals, low skeletal muscle mass may be more responsible for DRF than osteoporosis.

Introduction:

The Introduction Section is well written and gives a good introduction to the topic and leads the reader to the study’s research question.

Line 46 – 49: In my opinion, it would be fine to simply state the current definition of AWGS (2019). This section can be omitted to keep the paragraph focused.

->We omitted this section which about AWGS 2014

Line 55: a BMI of < 7.0 or < 5.4 cannot be the correct figure. In Ref. 7 these figures are referred to the ALM measurement

->Thank you for your careful reading, errors were excluded

Line 49-56: I would appreciate, if you could emphasize the 3 diagnostic groups “Muscle Strenght”, “Physical Performance” and “ASM” more. This would make this section more comprehensible.

->We emphasized the content by dividing it into three areas as you commented. Please find the revised manuscript below.

Low muscle strength as a handgrip strength of <28 kg for men and <18 kg for women;

Low physical performance as a 6-m walk of <1.0 m/s or short physical performance battery as a score of ≤9 or a 5-time chair stand test of ≥12 seconds;

Low appendicular skeletal muscle mass (ALM) as a <7.0 kg/m2 in men and <5.4 kg/m2 in women by dual-energy X-ray absorptiometry (DXA) or <7.0 kg/m2 in men and <5.7 kg/m2 in women by bioimpedance

Materials and Methods:

Thank you for providing the patient flowchart in Figure 1. It is a nice overview, how the number of patients was created.

->We applied propensity score matching method to reduce possible bias originated from the difference in patient’s demographic characteristics. The patients with DRF were matched with non-DRF group on a basis of a greedy algorithm of nearest neighbor matching at a 1:1 fixed ratio. Age was considered as predefined covariates. We obtained the same number of individuals for the DRF and non-DRF. Overall, 115 patients with DRF and 115 controls were enrolled before statistical comparison.

The statistical methods are appropriate for this study design.

Please include in the M&M Section, that you have assessed comorbidities. Moreover, please state, how you defined cardiovascular disease

->This study's definition of cardiovascular disease includes hypertension, coronary heart disease, peripheral arterial disease, and aortic disease. We added this sentence in the M and M section.

L 101: As you have assigned your participants to normal BMD, osteopenia and osteoporosis in Table 1, please specify “osteopenia” in this section.

->A definition of osteopenia has been added.

Line 103: Please introduce ALM before using it as abbreviation. Line 89 would be appropriate for doing so.

->We added introduction of ALM in introduction section

Results:

Table 1: The mean age in Non-DRF was stated as 71.45 ± 9.22. Did you also apply a max. age of 80 years for this group? Statistically, there must be some patients older than 80 with this mean and SD value

->In Table 1, Non – DRF was initially uncontrolled for age, resulting in 71.45 ± 9.22 values. The value can be over 80 years old in table 1 before matching. Since the age of non-DRF is matched with DRF through PSM, the age is matched with age in Table 2, so the control group does not reach 80 years old. 67.48 ± 8.32. In our medical center, almost surgery was not performed on DRF patients over the age of 80, and PSM was performed to match the age with the control group.

Discussion:

Line 162: I cannot find the information in the results section, where age was assessed as risk factor for DRF. Please clarify.

->Thank you for your careful reading, we rephrased this sentence.

Line 163-165: This sentence is rather confusion. Each variable resulting in a significant result in the multiple regression analysis should be stated equivalently. Please rephrase this sentence accordingly. ->We rephrased this sentence. Please find the revised manuscript below.

In our study, after propensity score matching was adjusted for age, women (OR 3.435, p=0.015), low skeletal muscle mass (OR 8.062, p=0.001), and CVD (OR 5.431, p<0.001) had a risk of DRF.

Line 166-172: Please state the information from this reference correctly. 86 should be rounded correctly to 87 years. “the same as the world's highest life expectancy in 2012, 169 with 57% of these being over 90 years” –> this sentence was reported in a confusing way. Please correct.

-> In order to focus on the part with greater relevance to the results of this study, that part has been omitted.

Line 173-174: I cannot find any statistical testing assessing a difference or correlation between age and sarcopenia in the results section.

->We omitted this section.

Line 184: As mentioned in the general comments, neither muscle strength nor physical performance were included as parameters in the present study. Thus, you cannot predict an association, because you did not assess one of the two parameters.

-> In order to focus on the part with greater relevance to the results of this study, that part has been omitted.

Line 196: “Coliskan et al concluded in their cross-sectional study including 27 patients with DRF and 28 controls without fractures that sacopenia…” would make this sentence more comprehensible.

->We rephrased this section. Please find the revised manuscript below.

Caliskan et al., a cross-sectional study involving 28 controls without fractures and 27 patients with DRF who visited geriatric outpatient clinics found that both groups' rates of sarcopenia were comparable.

Line 214: Do you mean that other studies investigation the association between sarcopenia and DRF did include younger patients. Please clarify.

->Yes, as you commented, there were studies conducted on relatively younger ages than ours.

Line 215: The precise definition of CVD must be included in the M&M Section. If available, you could include a detailed list of precise CVD in the results section.

->We added related information to M&M.

Line 228-231: I consider this sentence as very important in the discussion section. You may rephrase it in a way that DRF is considered as fracture type, which defines a “Severe or established osteoporosis”.  Please see

Kanis JA on behalf of the World Health Organization Scientific Group. Technical Report. World Health Organization Collaborating Centre for Metabolic Bone Diseases, University of Sheffield; UK: 2007. 2007. Assessment of osteoporosis at the primary health-care level

->Thanks for your comment, we rephrased this sentence. Please find the revised manuscript below.

Considering that DRF is considered as fracture type, which defines a “Severe or established osteoporosis”, the results of this study do not mean that the two factors are actually unrelated but that osteoporosis is common in the older population.

Line 249: Except that the dominant hand is normally slightly stronger, assessing the hand not affected by DRF could have been an option.

->You are right. Unfortunately, it was not measured due to the retrospective nature of this study. Thank you for the good comment. We will perform the next study as you commented.

Conclusion:

Line 258-259: It is hard to understand, what you want to express with this sentence. Please rephrase or omit, Line 259: why don’t you include female gender in this sentence as significant outcome of your multivariate regression analysis?

->We omitted & rephrased this sentence. Please find the revised manuscript below.

Although DRF is highly associated with osteoporosis, considering the above, factors related to general activity such as low skeletal muscle mass and CVD may be more important factors in causing DRF in women. In addition, research related to sarcopenia and DRF has been actively conducted, and future studies on sarcopenia will be of great assistance in the management of DRF, which is prevalent in society.

Reviewer 2 Report

This is an interesting study and well written, however, the cohorts are slightly different.

There is only one minor criticism: The non drf group had more men. Is it possible to do a further case match control study?

Author Response

This is an interesting study and well written, however, the cohorts are slightly different.

There is only one minor criticism: The non drf group had more men. Is it possible to do a further case match control study?

->Thank you for your careful reading and great comment. We will do a further case match control study in the future.

Reviewer 3 Report

Thank you for the opportunity to review this manuscript. I have some comments for the authors.

line 84: internal fixation is mentioned. Please give details regarding method (volar locking plate?)

I'm no expert on sarcopenia but the uso of DXA and lean bodymass to define sarcopenia has it´s limitations and does not follow the AWGS criteria. However, I believe the authors adress this properly in the discussion.

By using only DRFs that have undergone surgery with internal fixation, the fracturegroup has some bias. For example in elderly people with comorbidities and other health issues (like sarcopenia I presume), conservative treatment with a cast is often chosen due to limited functional demands. This is a limitation and could be commented on in the discussion.

Author Response

Thank you for the opportunity to review this manuscript. I have some comments for the authors.

line 84: internal fixation is mentioned. Please give details regarding method

->Thank you for great comment. All patients with DRF underwent open reduction and internal fixation surgery using 2.4mm Variable-Angle LCP Two-Column Volar Distal Radius Plate (Synthes, Oberdorf, Switzerland).

I'm no expert on sarcopenia but the uso of DXA and lean bodymass to define sarcopenia has it´s limitations and does not follow the AWGS criteria. However, I believe the authors adress this properly in the discussion.

->Thanks for your comment.

By using only DRFs that have undergone surgery with internal fixation, the fracture group has some bias. For example in elderly people with comorbidities and other health issues (like sarcopenia I presume), conservative treatment with a cast is often chosen due to limited functional demands. This is a limitation and could be commented on in the discussion.

->This study was conducted on patients who underwent surgery between 2015 and 2020. At that time, unfortunately, surgery was not decided based on the presence or absence of sarcopenia because of sarcopenia was relatively novel concept in these days.

However, it will be of great significance to compare the patients who underwent surgery and those who underwent conservative treatment in a future study. Thanks for your great inspiration.

We added this in the limitation section.

Round 2

Reviewer 1 Report

General Comments:

Thank you very much for the revised version of your manuscript.

However, I would have expected the authors to make a much higher effort to improve the previous version of the manuscript, especially considering that JCM is a Journal of high quality with an IF of 4.9. Principally, the font is still not consistent with the MDPI template. The different font size and line distance makes the manuscript hard to read. Although I recommended revising the discussion and conclusion section, I could only found some sentences or parts of sentences deleted, but no substantial modifications.

Although the methods and the data are principally interesting, the presentation lacks of adequate linguistic style as well as clarity and contains to many mistakes.  Thus, I cannot recommend acceptance of the present version of the manuscript.

Please find more detailed comments below:

Abstract:

Line 24: Again, you are not allowed to state that you investigated “sacropenia” in your study, because you did not assess all diagnostic parameters.

Line 30ff: I recommend stating that you found female gender, CVD and low skeletal muscle mass as significant predictors for DRF. “have risk of” is no good language style

Line 33: Why do you state “older, active individuals” in this section? Woman, CVD and low skeletal muscle were your significant parameters.

Introduction:

Line 51f: It is still unclear, how the AGS defines sarcopenia in this sentence: “sarcopenia should be described as low muscle mass plus low muscle strength and/or low physical performance”. Please see the flowchart in the cited reference. This is especially important, because you describe in Line 45 that not only muscle quantity (ALM), but also muscle quality is important for sarcopenia assessment.

Line 58: ALM is introduced as abbreviation in a confusion way. Either use “appendicular lean mass” (ALM) or “appendicular skeletal muscle mass” (ASM) throughout the whole manuscript.

Line 78: A verb is missing in the “which” clause. I recommend “is”.

M&M:

Line 86: “using” or “with” is missing before “2.4 mm VA-LCP…”

Line 93: This sentence does not include the right patient flow for the presents study. I recommend: “Data of individuals who underwent DXA for a health check up without any specific history of osteoporotic fracture (e.g., vertebral, hip, distal radius), which measures bone mineral density 93 (BMD) and appendicular lean body mass, were used as a patient collective for the control group.

Line 96: I would recommend using a paragraph in this section, because demographics were assessed in both DRF and non DRF cohort.

Line 108: If ALM was measured separately in the upper and lower extremities, in which body parts was the DXA for a health check up performed? In Table 1 you state simply T-scores of the Spine and Femur. Please clarify.

Results:

Line 125 f: I think the result of the propensity score-matching should belong in the Results section, where you mentioned it in the previous version of the manuscript.

Line 131 – 134: This section belongs in the M&M section, because they include the methodology or the include parameters of your study.

Line 150 – 152: it is not necessary to state the figures both in Table 3 and in the text. This information is redundant.

Line 153 / Table 3: “Non-low” is no good language style. “Normal” would be better.

Discussion:

Line 175: “patients with low skeletal muscle mass and CVD” would make more sense. CVD and skeletal muscle mass cannot be at risk for DRF.

Line 214f: You may have cited the reference in a wrong way. lower hip BMD, weaker grip strength, and lower appendicular mass in men were associated with the occurrence of DRF. However, grip strength and appendicular mass are diagnostic criteria for sarcopenia.

Line 219 – 222: It is not clear where you refer “this study” and “this is thought” to in these sentences. Please clarify.

Line 239: I think you mean “distal radius fracture” in this sentence.

Conclusion:

Line 268: This sentence would be more clear, if you state “Although DRF is highly associated with osteoporosis in the literature”

Author Response

Response to Reviewer 1 Comments

General Comments:

Thank you very much for the revised version of your manuscript.

However, I would have expected the authors to make a much higher effort to improve the previous version of the manuscript, especially considering that JCM is a Journal of high quality with an IF of 4.9. Principally, the font is still not consistent with the MDPI template. The different font size and line distance makes the manuscript hard to read. Although I recommended revising the discussion and conclusion section, I could only found some sentences or parts of sentences deleted, but no substantial modifications.

> Thank you very much for the sincere review on two occasions. In addition, we are sorry that we did not live up to the expectations of reviewer1 despite the numerous advices given to our paper. We reviewed our paper once again to match the MDPI template, and requested a paper language proofreading work to confirm it once more. In addition, the discussion and conclusion sections have been rewritten to convey the key points related to the results by focusing on the results of our research. Despite your busy schedule, we are sorry to ask you to review our thesis and thank you in advance.

Although the methods and the data are principally interesting, the presentation lacks of adequate linguistic style as well as clarity and contains to many mistakes.  Thus, I cannot recommend acceptance of the present version of the manuscript.

> We thoroughly reviewed our thesis as a whole and re-ordered the thesis language proofreading work.

Please find more detailed comments below:

Abstract:

Line 24: Again, you are not allowed to state that you investigated “sacropenia” in your study, because you did not assess all diagnostic parameters.

> Thank you for your comments. We rewritten the sentence.

New sentence : We aimed to investigate the correlation between DRF and low skeletal muscle mass which strongly correlated to sarcopenia

Line 30ff: I recommend stating that you found female gender, CVD and low skeletal muscle mass as significant predictors for DRF. “have risk of” is no good language style

> Thank you for your comments. We rewritten the sentence according to your comment.

New sentence : We found that female sex (odds ratio = 3.435, p=0.015), CVD (odds ratio = 5.431, p<0.001) and low skeletal muscle mass (odds ratio = 8.062, p=0.001) were significant predictors for DRF.

Line 33: Why do you state “older, active individuals” in this section? Woman, CVD and low skeletal muscle were your significant parameters.

> Considering that women in their late 60s are generally active, although they are older than the normal population, we used the expression "older, active indications" in line with the fact that our study's age group is in their 60s. However, since what you commented was correct, we changed our thoughts at first and rewritten the sentence by deleting the expression.

New sentence : Women with low skeletal muscle mass and CVD may be more responsible for DRF than osteoporosis.

Introduction:

Line 51f: It is still unclear, how the AGS defines sarcopenia in this sentence: “sarcopenia should be described as low muscle mass plus low muscle strength and/or low physical performance”. Please see the flowchart in the cited reference. This is especially important, because you describe in Line 45 that not only muscle quantity (ALM), but also muscle quality is important for sarcopenia assessment.

> According to the flow chart presented by AWGS 2019, the content has been rewritten in more detail from the initial process to the last process of sarcopenia diagnosis.

New sentence :

To diagnose sarcopenia in the primary health care or community preventive services setting, calf circumference (men: <34 cm, women: <33 cm) or SARC-F questionnaire (≥4) or SARC-F administration combined with CC measurement (SARC-CalF ≥11) was presented as the start of the study. Then, muscle strength measurement or physical performance was measured by the process of assessment, which measured handgrip strength (men: <28 kg, women: <18 kg), and physical performance measures 5-time chair stand test (≥12 s) before proceeding to the final diagnostic process. In the final diagnosis step, the following three were considered:

Low muscle strength as handgrip strength of <28 kg for men and <18 kg for women;

Low physical performance as a 6-m walk of <1.0 m/s or short physical performance battery score of ≤9 or a 5-time chair stand test of ≥12 seconds;

Low appendicular skeletal muscle mass (ASM) as a <7.0 kg/m2 in men and <5.4 kg/m2 in women by dual-energy X-ray absorptiometry (DXA) or <7.0 kg/m2 in men and <5.7 kg/m2 in women by bioimpedance

At this time, sarcopenia was defined as low ASM + low muscle strength or low physical performance, and severe sarcopenia was defined as low ASM + low muscle strength and low physical performance.

Line 58: ALM is introduced as abbreviation in a confusion way. Either use “appendicular lean mass” (ALM) or “appendicular skeletal muscle mass” (ASM) throughout the whole manuscript.

> All expressions have been changed to ASM to avoid confusion.

Line 78: A verb is missing in the “which” clause. I recommend “is”.

> Thank you for careful reading. Verb has been added as mentioned.

M&M:

Line 86: “using” or “with” is missing before “2.4 mm VA-LCP…”

> We apologize for the immature mistake. Added “using” as commented.

Line 93: This sentence does not include the right patient flow for the presents study. I recommend: “Data of individuals who underwent DXA for a health check up without any specific history of osteoporotic fracture (e.g., vertebral, hip, distal radius), which measures bone mineral density 93 (BMD) and appendicular lean body mass, were used as a patient collective for the control group.

> Thanks for your comment. Edited as suggested.

Line 96: I would recommend using a paragraph in this section, because demographics were assessed in both DRF and non DRF cohort.

> As you commented, we modified it using paragraphs.

Line 108: If ALM was measured separately in the upper and lower extremities, in which body parts was the DXA for a health check up performed? In Table 1 you state simply T-scores of the Spine and Femur. Please clarify.

> Thanks for the good question. DXA examination can measure bone density in the spine and femur as well as lean body mass in the upper and lower extremities. Appedicular lean body mass was calculated as the sum of soft tissue value except bone mineral content and fat mass of both arms and legs. Appendicular lean mass corresponds approximately to the ASM

Ref.

de Santana, F. M., D. S. Domiciano, M. A. Goncalves, L. G. Machado, C. P. Figueiredo, J. B. Lopes, V. F. Caparbo, L. Takayama, P. R. Menezes, and R. M. Pereira. "Association of Appendicular Lean Mass, and Subcutaneous and Visceral Adipose Tissue with Mortality in Older Brazilians: The Sao Paulo Ageing & Health Study." J Bone Miner Res 34, no. 7 (2019): 1264-74.

Lewandowicz, A., P. Slawinski, E. Kadalska, and T. Targowski. "Some Clarifications of Terminology May Facilitate Sarcopenia Assessment." Arch Med Sci 16, no. 1 (2020): 225-32.

Results:

Line 125 f: I think the result of the propensity score-matching should belong in the Results section, where you mentioned it in the previous version of the manuscript.

> That part has been moved to Results section

Line 131 – 134: This section belongs in the M&M section, because they include the methodology or the include parameters of your study.

> That part has been moved to M&M section

Line 150 – 152: it is not necessary to state the figures both in Table 3 and in the text. This information is redundant.

> That part has been deleted.

Line 153 / Table 3: “Non-low” is no good language style. “Normal” would be better.

> We revised it according to the comment.

Discussion:

Line 175: “patients with low skeletal muscle mass and CVD” would make more sense. CVD and skeletal muscle mass cannot be at risk for DRF.

> To make the sentence more natural to understand, we have modified it as you commented.

Line 214f: You may have cited the reference in a wrong way. lower hip BMD, weaker grip strength, and lower appendicular mass in men were associated with the occurrence of DRF. However, grip strength and appendicular mass are diagnostic criteria for sarcopenia.

> We deleted it because I thought it could cause confusion when the reader read it.

Line 219 – 222: It is not clear where you refer “this study” and “this is thought” to in these sentences. Please clarify.

> We deleted it because I thought it could cause confusion when the reader read it.

Line 239: I think you mean “distal radius fracture” in this sentence.

> We deleted it because I thought it could cause confusion when the reader read it.

Conclusion:

Line 268: This sentence would be more clear, if you state “Although DRF is highly associated with osteoporosis in the literature”

> We corrected it as you commented.

Thank you again for your detailed and careful review of our manuscript.
